# Preferences and Consumption of Pigeon Peas among Rural Households as Determinants for Developing Diversified Products for Sustainable Health

**Zahra Saidi Majili** [1,*], **Cornelio Nyaruhucha** [1], **Kissa Kulwa** [1], **Khamaldin Mutabazi** [2], **Constance Rybak** [3] and **Stefan Sieber** [3,4]

1   Department of Food Technology, Nutrition and Consumer Sciences, Sokoine University of Agriculture, Morogoro Box 3006, Tanzania; nyaruhu@suanet.ac.tz (C.N.); kkulwa@sua.ac.tz (K.K.)
2   School of Agricultural Economics and Business Studies, Sokoine University of Agriculture, Morogoro P.O. Box 3007, Tanzania; khamaldin2011@gmail.com
3   Leibniz Centre for Agricultural Landscape Research (ZALF e. V), Eberswalder Str. 84, 15374 Müncheberg, Germany; constance.rybak@zalf.de (C.R.); stefan.sieber@zalf.de (S.S.)
4   Department of Agricultural Economics, Faculty of Life Sciences Thaer-Institute, Humboldt-Universität zu Berlin, Unter den Linden 6, 10099 Berlin, Germany
*   Correspondence: majilizahra11@gmail.com

**Abstract:** Pigeon peas are legumes with a high nutritional value. Existing studies of pigeon peas in Tanzania mainly examine production and marketing, but little has been documented with respect to consumer preferences and the consumption of pigeon peas. This study assesses the preferences surrounding pigeon peas and their consumption as bases for the development of diversified and shelf-stable products for nutrition and income improvement. This study comprised 303 randomly selected farming households. Furthermore, 60 farmers participated in six focus group discussions in the Lindi region. A structured questionnaire and a checklist with guided questions were provided for data collection. The analysis uses SPSS (V.21), with differences between groups established using Kruskal–Wallis and Mann–Whitney tests. The associations were tested using Spearman's $\rho$ at $p < 0.05$. The mean pigeon peas consumption during the harvesting and lean seasons was 80 g/person/day and 18 g/person/day, respectively. The frequency of consumption was higher during the harvesting season (92%) than the lean (29%) season. The majority of farmers (91%) preferred to consume the local variety, with 84% of them consuming pigeon peas as stew. Five pigeon pea recipes exist in the area. The farmers identified availability, taste, source of income, and familiarity as the factors determining pigeon pea consumption and preferences. With limited recipes and other barriers limiting consumption, the creation of innovative ideas for the development of diversified and shelf-stable products fitting their consumption preferences is needed.

**Keywords:** pigeon peas; legumes; preference; consumption; Tanzania

## 1. Background

Consumption and demand for food are among the drivers of food production, which, in turn, exerts their influence on sustainability [1]. The sustainability of diets does not only include nutritional and environmental concerns, but also economic and socio-cultural dimensions [2]. It has been reported that some foods, such as vegetables and legumes, are healthy, as well as having a low environmental impact; hence, all these contribute more towards the goals of sustainability [3]. The pigeon pea is a dry mature legume seed of *Cajanus cajan L.*, from the family Fabaceae. It is widely grown in the developing world, including regions of Asia, Africa, Latin America, and the Caribbean [4]. It is mainly grown

in semi-arid tropical regions [5]. It is among the legumes that contribute towards food and nutrition security most significantly, hence contributing greatly to food sustainability in Sub-Saharan Africa.

Food sustainability involves a multitude of interrelated factors, including nutrition, environmental impacts, cultural preferences, safety, and food systems [6]. Adequate, safe, diversified, and nutrient-rich foods contribute to healthy diets; however, there are limitations posed by resource scarcity and environmental degradation, as well as unsustainable production, food losses, and unbalanced distribution and consumption patterns which influence consumers' diets [7]. The consumer behavior theory assumes that a consumer is a rational economic agent who aims to attain the highest possible satisfaction derived from affordable, nutritious, and safe food, as well as from its attributes (e.g., taste, color, and aroma) [8]. For a food product to be acceptable, consumers should identify a product that fits their preferences. Therefore, understanding consumer preferences and consumption behaviors is an important aspect in the designing of interventions related to sustainable diets, as well as the creation of a sustainable food system. This article focuses on presenting key findings that are related to consumer preferences and the consumption of pigeon peas as determinants for developing diversified and shelf-stable products for nutrition and income improvement. Considering consumer preferences and consumption behavior in product development will have a positive effect on physical and economic access to a variety of pigeon pea-based products that are adequate, culturally acceptable, and consumed sustainably.

In Sub-Saharan Africa, about 7.8 million households grow pigeon peas. According to the National Bureau of Statistics, in Tanzania, 209,299 households [9] and more than three-quarters of farmers in the southern zone grow pigeon peas [10,11]. Pigeon peas are rich sources of essential amino acids (lysine, methionine, and tryptophan), fiber, vitamins (riboflavin and niacin), and minerals (phosphorus, iron, and magnesium) [12,13]. Globally, it is estimated that about 4,982,000 tons of pigeon peas was consumed in 2015 [14], either as dehulled splits, whole, canned, boiled, roasted, or grind into flour to make a variety of desserts, noodles, snacks, and main dishes [12,15]. In Africa, it is estimated that 65% of pigeon peas produced are consumed by farmers [16,17]. The average consumption of pigeon peas in Sub-Saharan Africa is estimated to be around 0.4 kg/person/year [18]. In particular, Malawi has been reported to have the highest per capita consumption of pigeon peas (22.35 kg/year), followed by Kenya (6.72 kg/year) and Tanzania (5.16 kg/year) [14]. In Tanzania, pigeon peas are grown in several regions, including Manyara, Arusha, Lindi, Mtwara, Dodoma, Singida, Coastal, and Morogoro. Of the aforementioned 209,299 households that produce pigeon peas in Tanzania, 46,171 are from the Manyara region, followed by 40,405 in Lindi and 25,913 in Mtwara [9]. An average of 15,869 ha is cultivated in the Lindi region, of which 8971 ha are from the Nachingwea district and 4450 ha are from the Ruangwa district [9]. This study was conducted in the Lindi region, the second largest producer of pigeon peas in the country (NBS, 2012), where 80% of the households produce pigeon peas, contributing nearly 50% of Tanzania's total national production [11]. Although Manyara is the biggest producer in the country, its production is mainly for export purposes. Typically, pigeon peas produced in the Lindi region are used for household consumption and as a source of income, while in Manyara it is mainly a cash crop [19].

Despite the nutritional importance of pigeon peas, the crop is not adequately used for human consumption in Tanzania. It has been estimated that the per capita consumption of legumes in Tanzania is 14.14 g/d/person, which less than the 30 g/d/person recommended by the Food and Agriculture Organization of the United Nations (FAO) [20]. About 30%–35% of the produced pigeon peas were consumed as cooked green pigeon peas or dry peas [11,21]. Although Lindi is a high-pigeon-peas-producing area, it has a chronic malnutrition rate of 23.8% [22], as well as an anemia rate of 61% in children younger than 5 years [22] and of 32.5% in women of reproductive age [22]. Given that pigeon peas are good sources of amino acids and minerals and are affordable, combining them with other food groups will improve the quality of diet, hence reducing the chances of malnutrition.

Although existing studies of pigeon peas in Tanzania are mainly confined to production and marketing [11,23], there are a few studies on the consumption of pigeon peas [11,19,20,24] but no

information published on different recipes or shelf-stable products. Therefore, this study aims to (i) analyze existing recipes and consumption patterns related to consumer preferences, (ii) determine the nutritional knowledge and actual perception of pigeon pea consumption, and (iii) identify influential drivers and obstacles for their consumption in two villages. The results will act as a guide for developing diversified and shelf-stable products for nutrition and income improvement.

## 2. Methods

### 2.1. Study Design and Sample

A preference and consumption study was conducted in two semi-arid villages of Ruangwa (Mibure) and Nachingwea (Mitumbati) districts of Lindi region in October–December 2019 and March 2020. The two districts experience erratic, but adequate, rainfall between December and March, receiving an average of 400–800 mm rainfall per year with a 6.9% chance of precipitation. Despite this unpredictable rainfall, the two areas receive enough rainfall to grow pigeon peas. In these areas, pigeon peas are intercropped with maize. The two districts were selected because they are the leading producers of pigeon peas in the region and because of their varied market accessibility. The selection of villages was based on the high quantity of pigeon peas produced, based on information provided by the district agricultural office.

All adult males and females aged between 19 and 60 years, who grow pigeon peas on a small-scale level (i.e., ≤ 5 acres) in the selected villages, were eligible for this study. Fisher's formula [25], using the proportion of farmers who grow pigeon peas in the study area (80%), was used to calculate the desired sample size. A sample of 303 randomly selected farming households was chosen using the Microsoft Excel Random number function RAND. The lists of farming households were collected, with each household being assigned a unique number using the RAND function followed by the Microsoft Excel Ranking of numbers (RANK function) to generate values that were used to select households to be involved in the study.

The sample size for qualitative data was determined by the principles stipulated in Moser and Korstjens [26]. A total of n = 60 farmers were involved in focus group discussions to obtain insights and in-depth information on the preparation and cooking of pigeon peas as well as its consumption patterns and preferences. The permission for the study was granted by the Tanzania National Institute of Medical Research (NIMR) with reference number NIMR/HQ/R.8a/Vol. IX/3040. Written informed consent was obtained from each farmer before the interviews.

### 2.2. Data Collection

Data collection involved two sessions. During the first session, a household survey with face-to-face interviews was conducted at individual homesteads by trained interviewers. Using mobile tablets loaded with an open data kit tool for data collection, a structured, pretested questionnaire was employed. To ensure the data quality, constraints were loaded for impossible values and relevance for improbable ones. For example, the frequency of consumption should never be less than zero if a farmer reports consuming pigeon peas. Skip was added to remove unnecessary answers. Moreover, a Global Positioning System (GPS) was added to identify the data collection location.

#### 2.2.1. Household Characteristics

Information on age, gender, education level, marital status, income sources, and assets was collected to evaluate household characteristics. Household asset information was used to calculate the wealth index using factor analysis [27,28]. Fifteen assets, including toilet, water, bicycle, TV, radio, cell phone, hand hoe, rake, spade, axe, digging fork, motorcycle, cooking stove, tables, and chairs, were used to calculate the wealth index for each of the villages.

2.2.2. Preference and Consumption of Pigeon Peas

Information on the amount of pigeon peas consumed, consumption frequency, preferences, as well as knowledge and perception was also collected. The amount of pigeon peas consumed was collected using the 24 h food recall method [29]. Each interviewee was asked to mention all foods and amounts consumed in the past 24 h. Photos of household utensils were used to estimate the amount consumed and subjects were asked to indicate if the amount was consumed alone or shared to facilitate calculation of the amounts consumed per person per day. Subjects who did not consume pigeon peas during the 24 h prior to their interview received a follow-up phone call to make another appointment. A zero (0) amount was recorded if pigeon peas were not consumed in that particular week. Moreover, each interviewee was asked about the frequency of consumption of other legumes by reporting their usual consumption of each legume. The responses for their frequency of consumption were recorded in terms of the number of days in the week.

2.2.3. Nutritional Knowledge and Perceptions Surrounding Pigeon Pea Consumption

Nutritional knowledge and perceptions surrounding pigeon pea consumption were assessed using a three point Likert scale as a reduced scale from what is typically used by researchers [30]. The scale was reduced due to the nature of the study population, which involves respondents with a low level of education. The value of 1 stands for disagree/not acceptable, 2 stands for neither agree nor disagree, and 3 stands for agree/acceptable. A total of fourteen (14) questions related to knowledge and perception of pigeon peas attributes were asked.

2.2.4. Qualitative Information

The second session collected qualitative information for the contextualization of the research question through a Focus Group Discussion (FGDs) guided by a checklist. Information on pigeon pea preparation, consumption, and preference was collected. Six focus group discussions were conducted at the village centre, with each village represented by n = 30 farmers, including both males (15) and females (15). In each village, the discussions involved three different groups with an average of n = 10 farmers in each group. The first two groups were gender specific (i.e., either male or female) and the third group included both male and female farmers.

*2.3. Data Analysis*

Statistical analyses were completed using Statistical Product and Service Solutions (SPSS) software version 20. The Shapiro–Wilk test was used to check data normality prior to analysis. During the analysis, all assets variables were changed to binary. International standards were used to categorize information on water and toilets into binary variables [31]. Frequency was analyzed to check the acceptability of each variable. The asset was included in the analysis if the frequency was between 5% and 95%, as a percentage below 5 is considered very rare, while those greater than 95% are considered more common; both extremes are unable to differentiate farmers using the wealth index. Factor analysis was done to compute factor weights, means, and standard deviations for each household asset. Means and standard deviations for each household asset were used for the standardization of assets' data, followed by multiplication of the factor weights to obtain wealth scores. The wealth scores were ranked using the Rank case command in SPSS to rank and assign each household to one of the wealth quintiles from the poorest to the wealthiest [28]. Furthermore, knowledge and perception scores were changed into binary variables before summing to get separate knowledge and perception scores. A score of one (1) was given for a correct response and zero (0) for an incorrect response related to knowledge. Similarly, the scores were given if the respondent agreed (score = 1) or disagreed (score 0) on perceived attitude. The score for both knowledge and perception were then summarized to get meaningful information.

Means and standard deviations were used to summarize continuous variables (age, household size) and frequencies to summarize categorical variables (variety of pigeon peas consumed, pigeon peas based dishes, consumption frequencies, preferences, as well as knowledge and perception scores). Significance tests were computed using the Mann–Whitney U test for comparing categorical variables with two groups (i.e., gender, age, agricultural season, and household heads) against pigeon pea consumption frequency and preference. The Kruskal–Wallis test was used to compare consumption frequency and preferences with categorical variables for three groups, including marital status, education level, occupation, wealth quintiles, and factors influencing their preferences. Moreover, the Spearman correlation was computed to determine the associations in frequencies of consumption and preference of various pigeon peas dishes against the farmers' knowledge, perception, sensory attributes, and drivers for their choice. Multiple linear regressions were computed to determine factors that influence the consumption of pigeon peas. The model was fitted against the dependent variable (frequency of consumption of pigeon peas) and the independent variables (availability, affordability, nutrition knowledge, accessibility, preparation time, and taste). The statistical significance was considered at *p* value < 0.05. Deductive thematic content analysis was used to summarize themes and factors for qualitative information by using a matrix table.

## 3. Results

### 3.1. Household Characteristics

The mean age of respondent farmers was 35.8 ± 8.5(SD), with the majority aged between 15 and 49 years (Table 1). The mean household size was 3.5 ± 1.5(SD) and 80% of the households had a male head. In terms of wealth quintiles, 40% of farming households were poor, including 45% of those from Mibure and 34% from Mitumbati villages (Table 1).

**Table 1.** Household characteristics.

| Variables | Overall (n = 303) | | Mibure (n = 152) | | Mitumbati (n = 151) | |
|---|---|---|---|---|---|---|
| | n | % | n | % | n | % |
| **Age** | | | | | | |
| 15–49 years | 289 | 95 | 148 | 97 | 141 | 93 |
| >49 years | 14 | 5 | 4 | 3 | 10 | 7 |
| **Gender** | | | | | | |
| Male | 186 | 61 | 97 | 64 | 89 | 59 |
| Female | 117 | 39 | 55 | 36 | 62 | 41 |
| **Household heads** | | | | | | |
| Female headed household | 62 | 20 | 31 | 20 | 31 | 21 |
| Male headed household | 241 | 80 | 121 | 80 | 120 | 80 |
| **Marital status** | | | | | | |
| Married | 215 | 71 | 104 | 68 | 111 | 74 |
| Divorced | 43 | 14 | 25 | 16 | 18 | 12 |
| Single | 39 | 13 | 21 | 14 | 18 | 12 |
| Widowed | 6 | 2 | 2 | 1 | 4 | 3 |
| **Education level** | | | | | | |
| No formal education | 42 | 14 | 31 | 20 | 11 | 7 |
| Primary school education | 259 | 85 | 121 | 80 | 138 | 92 |
| Secondary education or higher | 2 | 1 | 0 | 0 | 2 | 1 |
| **Occupation** | | | | | | |
| Farmer | 292 | 77 | 150 | 77 | 142 | 76 |
| Employed in the informal sector (casual labour) | 18 | 5 | 9 | 5 | 9 | 5 |
| Self employed | 70 | 18 | 34 | 19 | 36 | 19 |
| **Household wealth quintile** | | | | | | |
| Poorest | 74 | 24 | 42 | 28 | 32 | 21 |
| Middle | 153 | 51 | 74 | 49 | 79 | 52 |
| Wealthiest | 76 | 25 | 36 | 24 | 40 | 27 |

*3.2. Pigeon Peas Consumption*

3.2.1. Existing Pigeon Peas Recipes: Preparation and Cooking Methods

Five different ways of preparing and cooking pigeon peas dishes were identified in the study area during FGDs (Table 2). All group members in the six FGDs reported preparing pigeon peas in different dishes including stew from whole pigeon pea grains (relish) accompanied by rice or a stiff porridge. There were also reports of boiling green pigeon pea pods to be eaten as a snack, mixed with other foods, or cooked as the main dish. About 83% of farmers reported preparing pigeon peas as a stew cooked with green pigeon peas, dried pigeon peas, or pigeon peas splits (dhal).

**Table 2.** Existing pigeon pea recipes: preparation and cooking methods.

| Themes | Subthemes | Preparation Method | Response | |
|---|---|---|---|---|
| | | | n | % |
| Dishes consumed | | We consume pigeon peas in several ways, namely dried pigeon peas stew (DPPS), Green pigeon peas stew (GPPS), Dhal stew (DS), snack ("*mikumbu*"), main dish (MD), and mixed with another food (MPPF). | 60 | 100 |
| Existing recipes (Cooking method and preparation) | GPPS and DPPS | For green and dried pigeon peas, we peel, wash, and boil until well cooked. Then, we partially fry onions and tomatoes, before adding boiled pigeon peas, salt, and some water to get stew. | 50 | 83 |
| | | For green and dried pigeon peas, we peel, wash, and boil until well cooked. After boiling, we add onion, tomato, salt, and coconut milk to get a stew that is consumed with rice or stiff porridge | 40 | 67 |
| | Snack | We usually boil green pigeon peas with their pods and consume it as a snack while preparing the meals. This is mostly given to children to reduce hunger while we prepare the main meal. | 40 | 67 |
| | DS | We roast dried pigeon peas in the ashes then grind it in mortar to remove the husk and then grind it with stones to get small split. These splits are then boiled and relished with onion, salt, tomato. | 50 | 83 |
| | MPPF | We also consume dried pigeon peas, which we boil with dehulled maize and relish with oil, coconut milk, or sesame milk before consuming it as a main dish ("*makande*")Dried pigeon peas are boiled and mixed with cassava or sweet potatoes, then consumed as the main dish; however, it is rarely prepared in this way. | 35 | 58 |
| | MD | Sometimes we boil dried pigeon peas and relish it with salt alone or with salt and coconut milk, then consume it as the main dish with porridge. | 43 | 72 |

3.2.2. Frequency of Consumption of Pigeon Peas

The majority of farmers consume pigeon peas within a week during the harvesting (280; 92%) and lean (90; 29%) seasons. The mean intake of pigeon peas during the harvesting season is 80 g/person/day,

but only 18 g/person/day during the lean season. The results regarding pigeon pea consumption in terms of residence, agricultural season, and household characteristics are presented in Table 3. During harvesting, 44% of the farmers consumed pigeon peas more than five days in a week, but only 4% do so during the lean season. In terms of residence, 55% of farmers in Mitumbati and 32% in Mibure consume pigeon peas more than five days a week. There is a significant difference in the consumption of pigeon peas across agricultural seasons, area of residence, and source of income. Furthermore, those dependent on farming activities (45%) consume pigeon peas more frequently than those who were self-employed or depend on the informal sector. There is no enough evidence to determine significant differences in terms of age, education level, marital status, head of the households, or wealth tertiles (Table 3).

**Table 3.** Frequency of consumption of pigeon peas.

| Household Characteristics | Consumed 1–3 d/w | | Consumed 4–5 d/w | | Consumed >5 d/w | | *p*-Value |
|---|---|---|---|---|---|---|---|
| | n | % | n | % | n | % | |
| **Village** [a] | | | | | | | 0.000 * |
| Mibure | 68 | 45 | 35 | 23 | 49 | 32 | |
| Mitumbati | 44 | 29 | 24 | 16 | 83 | 55 | |
| **Agricultural season** [a] | | | | | | | |
| Harvest season | 89 | 29 | 59 | 20 | 132 | 44 | 0.000 * |
| Lean season | 68 | 22 | 18 | 6 | 4 | 1 | |
| **Age** [a] | | | | | | | |
| 15–49 years | 109 | 38 | 57 | 20 | 123 | 43 | 0.181 |
| >49 years | 3 | 21 | 2 | 14 | 9 | 64 | |
| **Gender** [a] | | | | | | | |
| Male | 62 | 33 | 43 | 23 | 81 | 44 | 0.395 |
| Female | 50 | 43 | 16 | 14 | 51 | 43 | |
| **Marital status** [a] | | | | | | | |
| Married | 61 | 69 | 46 | 78 | 92 | 70 | |
| Divorced/Single/Widowed | 28 | 31 | 13 | 22 | 40 | 30 | 0.949 |
| **Education level** [b] | | | | | | | |
| No formal education | 16 | 37 | 7 | 16 | 20 | 47 | |
| Primary school education | 92 | 38 | 45 | 19 | 104 | 43 | 0.735 |
| Secondary education or higher | 4 | 21 | 7 | 37 | 8 | 42 | |
| **Occupation** [a] | | | | | | | |
| Agriculture | 8 | 91 | 59 | 100.0 | 130 | 99 | |
| More than agriculture | 8 | 9 | 0 | 0 | 2 | 1 | 0.017 * |
| **Household heads** [a] | | | | | | | |
| Female headed household | 22 | 36 | 7 | 11 | 33 | 53 | 0.392 |
| Male headed household | 90 | 37 | 52 | 22 | 99 | 41 | |
| **Household wealth quintile** [b] | | | | | | | |
| Poorest | 21 | 23 | 15 | 25 | 31 | 23 | |
| Middle | 39 | 44 | 32 | 55 | 72 | 55 | 0.218 |
| Wealthiest | 29 | 33 | 12 | 20 | 29 | 22 | |

**d/w = days per week.** [a] Mann–Whitney U test for two categorical groups (e.g., yes/no), [b] Kruskal–Wallis test for more than two categories. * Significant at *p* < 0.05.

In terms of the consumption of different pigeon pea dishes, the results indicate significant differences regarding the consumption of pigeon pea dishes during the harvesting season and the lean season. Specifically, the farmers consume pigeon pea stews 1–3 days in a week during the harvesting (55%) and the lean (46%) seasons. Only 16% of the farmers consume pigeon peas stew more than 5 days in a week (Figure 1).

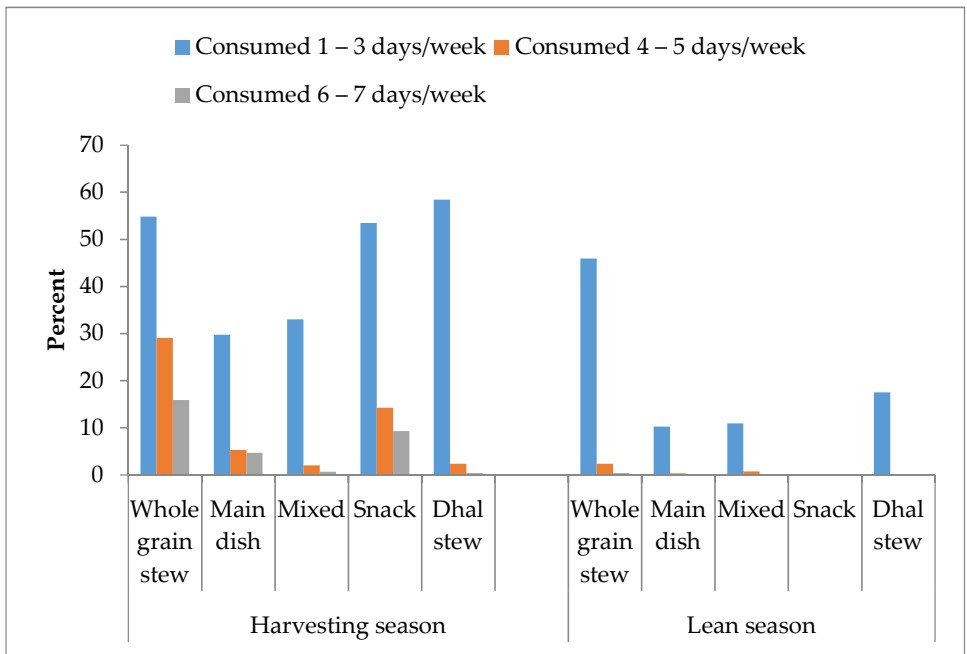

**Figure 1.** Consumption of pigeon peas dishes during harvesting season and lean season.

Furthermore, the majority (100%) of the FGD members reported consuming pigeon peas more frequently when green during the harvesting season not only because of its taste but also because they cause less flatulence. Furthermore, the short maturity period is reported as a factor for consuming green pigeon peas during the harvesting season. The respondents reported the consumption of dried pigeon peas as a relish (92%) to their main meal as well as the consumption of them, mixed with maize to obtain the maize pigeon pea dish ("Kande") or mixed with cassava/sweet potatoes ("Futari"). The consumption of pigeon peas during the lean season (50%) is reported to decrease as members claim that, during this time, people are preparing their farms and no green pigeon peas are available. The consumption of dried pigeon peas (62%) observed is due to their availability during the time of the year when people have limited funds: people usually consume what is available at home and pigeon peas are the main legume that almost every household keeps in stock.

*3.3. Pigeon Peas Consumption Preference*

Farmers (91%) prefer consuming local varieties of pigeon peas, with only 8% preferring to consume the improved variety. In terms of dishes, 84% of farmers preferred to consume stews, among them: 47% preferred to consume whole dried pigeon peas, 21% green pigeon peas, and 16% dhal stews. Table 4 indicates the consumption preferences for different pigeon peas based dishes. It is observed that more than 50% of farmers in Mitumbati consumed more than three dishes, but only one dish in Mibure was consumed. The results on the other hand reveal that there is a significant difference in consumer preferences and the consumption of pigeon pea dishes among education level and household heads (Table 4). On the other hand, 100% of the focus group discussion members reported preferring to consume pigeon pea stew either green or dried. They also wished to learn how to store green pigeon peas so that it can be consumed throughout the year. The reason for the choice was that green pigeon peas are tastier than dried ones, which require a lot of spices in order to taste good. Likewise, they

report that dried pigeon peas are easily attacked by pests. Furthermore, dried pigeon peas must be dehulled to get dhal before cooking; this is a long process as they use stones to process it. The FGD members (87%) stated that the familiarity of a consumer's behavior is one of the reasons for their consumption preferences of pigeon peas stew.

**Table 4.** Preference for consumption of pigeon-pea-based dishes.

| Household Characteristics | Whole Grain Stew | | Green Pigeon Peas Stew | | As Dhal Stew | | As Snack | | Main Dish | | *p*-Value |
|---|---|---|---|---|---|---|---|---|---|---|---|
| | n | % | n | % | n | % | n | % | n | % | |
| **Villages [a]** | | | | | | | | | | | |
| Mibure | 63 | 44 | 36 | 56 | 25 | 52 | 18 | 60 | 10 | 56 | 0.059 |
| Mitumbati | 80 | 56 | 28 | 44 | 23 | 48 | 12 | 40 | 8 | 44 | |
| **Age [a]** | | | | | | | | | | | |
| 15–49 years | 136 | 95 | 63 | 98 | 45 | 94 | 29 | 97 | 16 | 89 | 0.764 |
| >49 years | 7 | 5 | 1 | 2 | 3 | 6 | 1 | 3 | 2 | 11 | |
| **Gender [a]** | | | | | | | | | | | |
| Male | 86 | 60 | 43 | 67 | 26 | 54 | 18 | 60 | 13 | 72 | 0.793 |
| Female | 57 | 40 | 21 | 33 | 22 | 46 | 12 | 40 | 5 | 28 | |
| **Marital status [b]** | | | | | | | | | | | |
| Married/cohabitating | 90 | 63 | 47 | 73 | 35 | 73 | 26 | 87 | 17 | 94 | 0.725 |
| Single/Divorced/widowed | 53 | 37 | 17 | 27 | 13 | 27 | 4 | 13 | 1 | 6 | |
| **Education level [b]** | | | | | | | | | | | |
| No formal education | 12 | 8 | 13 | 20 | 7 | 15 | 26 | 87 | 3 | 17 | |
| Primary school education | 131 | 92 | 50 | 78 | 41 | 85 | 3 | 10 | 14 | 78 | 0.020 * |
| Secondary education or higher | 0 | 0 | 1 | 2 | 0 | 0 | 1 | 3 | 1 | 6 | |
| **Occupation [b]** | | | | | | | | | | | |
| Farmer | 135 | 78 | 64 | 75 | 46 | 75 | 29 | 76 | 18 | 86 | |
| Employed in informal sector (casual labour) | 6 | 4 | 3 | 4 | 5 | 8 | 3 | 8 | 1 | 5 | 0.176 |
| Self-employed (small business) | 32 | 18 | 18 | 21 | 10 | 16 | 6 | 16 | 2 | 10 | |
| **Household heads [a]** | | | | | | | | | | | |
| Female headed household | 37 | 26 | 9 | 14 | 13 | 27 | 2 | 7 | 1 | 6 | 0.021 * |
| Male headed household | 106 | 74 | 55 | 86 | 35 | 73 | 28 | 93 | 17 | 94 | |

[a] Mann–Whitney U test for two categorical groups (e.g., yes/no), [b] Kruskal–Wallis test for more than two categories, * Significant at $p < 0.05$.

### 3.4. Farmer's Nutritional Knowledge and Perception of Consumer Preferences and the Consumption of Pigeon Peas

Table 5 indicates the farmer's knowledge with regards to their preferences and consumption of pigeon peas. Farming household respondents (37%) agreed that the pigeon peas are an important source of protein for their families and 93% perceived pigeon peas to have a good taste. In terms of consumption preferences surrounding pigeon peas, a significant association was observed for good taste ($r_s = 0.113$, $p = 0.049$) (Table 5).

**Table 5.** Farmer's knowledge and perception on consumer preference and consumption of pigeon peas.

| | Agree | | Consumption Frequency | | Consumption Preference | |
|---|---|---|---|---|---|---|
| **Knowledge Tested** | **n** | **%** | **$r_s$** | **$p$-Value** | **$r_s$** | **$p$-Value** |
| Rich in protein | 13 | 4 | 0.109 | 0.059 | 0.003 | 0.957 |
| Rich in iron | 18 | 6 | 0.013 | 0.823 | −0.004 | 0.940 |
| Rich in micronutrients | 21 | 7 | −0.047 | 0.411 | −0.045 | 0.437 |
| Not rich in energy | 260 | 86 | 0.044 | 0.442 | −0.039 | 0.500 |
| Correct serving size | 74 | 24 | −0.102 | 0.077 | −0.100 | 0.083 |
| Pigeon peas are the important source of protein to your family | 113 | 37 | −0.039 | 0.501 | −0.045 | 0.440 |
| Children are taught about importance of pigeon peas | 114 | 38 | −0.014 | 0.808 | −0.070 | 0.224 |
| **Perceived attributes of pigeon peas** | | | | | | |
| Pigeon peas have a good taste | 283 | 93 | 0.031 | 0.590 | 0113 | 0.049* |
| Pigeon peas are source of income | 194 | 64 | 0.060 | 0.298 | 0.136 | 0.018* |
| Colour | 3 | 15 | 0.002 | 0.973 | 0.024 | 0.679 |
| Texture | 3 | 15 | −0.017 | 0.773 | −0.050 | 0.384 |
| Flavour | 4 | 20 | 0.027 | 0.638 | −0.031 | 0.589 |
| Size | 4 | 20 | 0.010 | 0.860 | 0.009 | 0.873 |
| Aroma | 6 | 30 | 0.008 | 0.884 | 0.003 | 0.960 |

* Spearman's correlation coefficient ($r_s$) is significant at 0.05 levels.

### 3.4.1. Drivers for Consumption of Pigeon Peas

Availability (78%) and taste (46%) are among the drivers for the consumption frequency and preferences (Table 6). Spearman's correlation coefficient ($r_s$) indicates that there is a significant association between the consumption frequency of pigeon peas and household preferences ($r_s = 0.122$, $p = 0.034$). It is also observed that the consumption preference for pigeon peas is associated with the availability of pigeon peas ($r_s = 0.261$, $p = 0.000$) and familiarity ($r_s = 0.120$, $p = 0.036$).

**Table 6.** Factors to consider when choosing to eat pigeon peas.

| | Agree | | Consumption Frequency | | Consumption Preference | |
|---|---|---|---|---|---|---|
| **Factors** | **n** | **%** | **$r_s$** | **$p$-Value** | **$r_s$** | **$p$-Value** |
| Taste | 139 | 46 | 0.024 | 0.675 | 0.064 | 0.265 |
| Quantity | 23 | 8 | −0.008 | 0.892 | −0.089 | 0.123 |
| Availability | 237 | 78 | −0.026 | 0.652 | 0.261 | 0.000 ** |
| Price | 28 | 9 | 0.065 | 0.263 | −0.069 | 0.229 |
| Psychological factors (familiarity) | 28 | 9 | 0.006 | 0.923 | 0.120 | 0.036 * |
| Social | 9 | 3 | 0.004 | 0.951 | −0.065 | 0.260 |
| Shelf life | 14 | 5 | −0.016 | 0.785 | −0.065 | 0.263 |
| Preference of the household | 51 | 17 | 0.122 | 0.034 * | −0.083 | 0.151 |

* Spearman's correlation coefficient ($r_s$) is significant at 0.05 levels, ** Spearman's correlation coefficient ($r_s$) is significant at 0.01 levels.

### 3.4.2. Factors Influencing the Consumption of Pigeon Peas

A multiple regression model was statistically significant, predicting the consumption of pigeon peas during harvesting season: $F_{(11, 292)} = 1.769$, $p (0.035) < 0.05$. The coefficient of determination ($R^2$) explained 6.8% of the variability of dependent variable. Table 7 indicates variables included in the model. Out of the eleven variables, the availability of pigeon peas in the area significantly influence the consumption of pigeon peas ($p = 0.10$).

**Table 7.** Multiple linear regression model predict consumption of pigeon peas.

| Factors | Harvesting Season | | | Lean Season | | |
|---|---|---|---|---|---|---|
| | **B** | **t** | ***p* Value** | **B** | **t** | ***p* Value** |
| Constant | 1.611 | 0.899 | 0.370 | 0.447 | 0.384 | 0.702 |
| Age | 0.638 | 1.001 | 0.318 | 0.340 | 0.820 | 0.413 |
| Gender | −0.250 | −0.867 | 0.387 | −0.045 | −0.237 | 0.813 |
| Marital status | 0.003 | 0.011 | 0.991 | 0.064 | 0.320 | 0.749 |
| Education | 0.017 | 0.051 | 0.959 | 0.289 | 1.364 | 0.174 |
| Occupation | 0.829 | 1.146 | 0.253 | −0.322 | −0.685 | 0.494 |
| Availability | 0.779 | 2.214 | 0.028* | 0.164 | 0.716 | 0.475 |
| Affordability/price | 0.635 | 1.451 | 0.148 | −0.207 | −0.727 | 0.468 |
| Nutrition Knowledge | −0.836 | −1.089 | 0.277 | 0.464 | 0.929 | 0.354 |
| Accessibility | 0.390 | 1.176 | 0.241 | 0.044 | 0.205 | 0.837 |
| Preparation time | 0.011 | 0.016 | 0.987 | −0.184 | −0.413 | 0.680 |
| Good taste | 0.408 | 1.465 | 0.144 | −0.192 | −1.063 | 0.289 |
| F- statistic of the model | $F_{(11,292)} = 1.769$ | | | $F_{(11,292)} = 0.556$ | | |
| Coefficient of determination ($R^2$) | 6.8% | | | 2.1% | | |
| Significance of the model (*p*-value) | 0.035 | | | 0.863 | | |

\* Significant at $p < 0.05$.

## 4. Discussion

### 4.1. Pigeon Peas Consumption

Pigeon pea is a semi-arid tropical legume that is rich in protein and micronutrients. It is widely used as an affordable source of protein. In the Lindi region, pigeon peas are used for both household consumption and as a source of income. The findings of the study indicate that the amount of pigeon peas consumed during harvesting season is greater than that recommended by the FAO for legumes consumption (30 g/person/day). The high frequency of pigeon peas consumption observed during harvesting season could be due to their high availability, as almost all households in the study area grow pigeon peas. On the contrary, the situation is different during the lean season, when the mean intake of pigeon peas and all legumes drop to 18 g/person/day and 20 g/person/day, respectively. These values are less than the FAO recommendations. The low amount of the consumed pigeon peas is due to their unavailability caused by inadequate storage and processing techniques as well as the dependency of agricultural activities on rainfall. This situation affects the sustainable consumption of pigeon peas as a nutritious and affordable legume in the study area. According to Szczebyło and colleagues, increasing the consumption of pulses constitutes an important component of the dietary shift toward more sustainable and healthy diets [32].

Furthermore, less diversified recipes exist in the study area. It is found that pigeon peas are mostly consumed as a stew made from green, dried, and dehulled splits of pigeon peas and limited other forms of consumption. This limits the frequency of consumption of pigeon peas due to their monotonous taste, which is among the determinants underlying their consumption. Worldwide, pigeon peas can be used in a variety of recipes, thus increasing the quality and organoleptic properties of pigeon peas [12,15,33–36] and increasing the frequency of their consumption. The observed cooking preparations (recipes) are due to limited knowledge on how to prepare pigeon peas in different ways owing to limited exposure to different preparation techniques. The lack of knowledge surrounding legumes' preparation and the time involved in this preparation is reported by Figueira and colleagues as limiting factors for the consumption of legumes [37]. Hence, increasing the skills and techniques regarding the preparation of pigeon peas into diversified products could reduce preparation and cooking time. It would also

widen culinary attribute choices and increase the frequency of consumption. Doing so would mean that Tanzanians could sustainably consume the recommended amount of legumes year round.

The frequency of the consumption of pigeon peas decreases during the lean season for all kinds of dishes. The proportion of farmers consumed pigeon peas, both whole seed and dhal stew, decreased by 4% and 48% during the lean season, respectively. The reason for the low frequency of consumption could be due to low grain yields because a significant amount is consumed while green due to high post-harvest losses. Other researchers [38,39] report significantly low yields due to consumption of green peas as well as pest infestations affecting the quality of pigeon peas grain. A large decrease in the consumption of dhal stew is due to limited availability of time to prepare dhal, as reported during focus group discussions. Dhal is prepared locally, using a traditional grinding stone to make the splits after having been roasted for some time. This is time consuming, and, hence, farmers opt for other dishes as, during harvest time, farmers are busy with farm work and they are unable to stay at home. Additionally, the existing processing capacity among farming household hinders the frequency of consumption of pigeon peas due to inadequate storage capacity and poor processing technology. The barrier for consumption of dhal due to perceived time for preparation and the use of stone for processing is an opportunity for promoting innovative processing technologies that ensure availability in large quantities as well as reducing postharvest losses. The dependency on the rainy season and a lack of irrigation schemes in the study area contribute to the decreased frequency of consumption of pigeon peas in the form of green boiled pods (snack) during the lean season. This limits the availability of green pigeon peas, which are mostly used for snacking and preferred for cooking as stews. Thus, promoting home gardening could increase the availability of pigeon peas.

### 4.2. Consumption Preferences of Pigeon Peas

It was found that farmers in the study area preferred to consume pigeon peas as a stew and as a snack (boiled green pods). The reason for their preference is a learning experience (familiarity), as they grew up consuming pigeon peas in these ways, as reported during focus group discussions. This is also observed during the survey, where people reported consuming pigeon peas in a way similar to their elders. Thus, the taste is familiar. Similar findings are reported by Vabø and Hansen [40] as well as Monge and colleagues [41]. According to Lipsky and colleagues, people may prefer to eat certain food due to what is available in their environment [42]. Similar behavior is observed in the study area, where more than 75% of the households grow pigeon peas on their farms or around their homestead. This makes pigeon peas more readily available than other legumes, which are either grown in very small quantities or not grown at all due to climatic conditions. Hence, this makes other legumes more expensive than what they grow themselves.

In terms of varieties, both survey results and focus group discussions reveal that local or traditional pigeon peas are preferred due to their taste, availability, and resilience against pests. This is consistent with the findings of Dalton and Regier [23]. Moreover, the preference of the consumption of pigeon pea dishes differs significantly with education level and household heads. Those with primary education preferred to consume pigeon peas dishes more frequently than those with other education levels. This is because the majority of them depend on farming, and pigeon peas are among the leguminous crops grown in the area, hence making them available. This is different from the results of the previous studies, which found that it is the well-educated individuals who consumed more pigeon peas [20,43]. It is also observed that farmers consume pigeon peas due to their taste and familiarity, not because of their nutritional benefits. Therefore, educating families on the nutritional benefits of pigeon peas could increase consumer preferences for, and the frequency of consumption of, pigeon peas for health reasons.

### 4.3. Knowledge and Perception about Pigeon Peas Consumption

Farmers in the study area know little about the importance of pigeon peas for consumption. This could be due to the limited nutrition education they have with regard to healthy eating. The majority of respondents have only completed primary education, where little is taught about the importance of

nutritious and diversified diets. Thus, children entering adulthood are not educated about good eating habits and diverse diets. However, there is a nutrition education program in the study area provided through health centers. Unfortunately attendance is limited, mainly by those seeking reproductive and child health services. The education provided in these centers mostly focuses on maternal and infant feeding, with little given on the nutritional well-being of other groups, especially those not of productive ages. Thus, the majority of the farmers does not have general nutritional knowledge or understand the importance of consuming different food groups, including legumes. This affects their consumption patterns, hence leading to a poor nutritional status that could result in lower labor productivity. Ultimately, this increases food and nutrition insecurity in the community. The results also indicate significant differences in terms of perception, consumption frequency, and preferences of pigeon peas. The farmers in the study area tend to consume what is readily available; this is also evident from the differing frequency of pigeon pea consumption across harvesting and lean seasons. During the lean season, a limited amount of pigeon peas is available; hence, it is consumed less frequently. Furthermore, occupation is among the observed determinants for the consumption of pigeon peas. The majority of farmers in the study area practice a subsistence way of farming; hence, they have little income. This hinders the consumption of other protein-rich foods that are more expensive. Pigeon peas are also a source of income in the community, if the market value of grain pigeon peas was high, it would negatively affect consumption. Hence, diversifying pigeon peas into different products will promote the use of pigeon peas within the country, consequently increasing their use as a source of income. This will promote greater production of pigeon peas as they improve soil fertility, which in turn contributes to environmental and agricultural sustainability. In addition, pigeon peas have the ability to bring minerals from deep soil horizons surface and hence improving soil air circulation [4]. Moreover, they have the ability to maintain photosynthetic function during stress compared to other legumes. Hence, promoting the consumption of pigeon peas will create more demand, resulting in more production and increased agricultural sustainability.

## 5. Conclusions and Recommendations

In the study area, pigeon peas are among the most important legumes for helping families to consume the recommended amounts of nutrients. However, limited recipes and knowledge of how to prepare pigeon peas, along with poor nutrition education, inadequate storage and processing techniques, social behavior learning as well as rainfall dependency and the use of stones for processing dhal are among the barriers for pigeon peas consumption. All these necessitate not just the need to develop new recipes and provide cooking demonstrations but also to conduct research that finds innovative ideas for the development of diversified and shelf-stable products that improve nutrition and income. Additionally, nutrition education should be implemented. Its design should foster on promoting healthy eating to all age group, thus improving the preference for, and consumption of, pigeon peas and other food groups throughout the year.

**Author Contributions:** Z.S.M. countersigned on the design of the study, data collection, and performed the statistical analysis; as well as wrote the first draft of the manuscript. Other authors (C.N., K.K., K.M., C.R. and S.S.) critically reviewed and refined the manuscript. All authors have read and agreed to the published version of the manuscript.

**Funding:** This research was funded by Federal Ministry of Food and Agriculture (BMEL) based on a decision of the Parliament of the Federal Republic of Germany via the Federal Office for Agriculture and Food (BLE). Grant number Vegi_Leg/2816PROC09/24.08.2018. No funds provided for APC.

**Acknowledgments:** The authors acknowledge the financial support from the Vegi Leg project. The project is supported by funds from the Federal Ministry of Food and Agriculture (BMEL) based on a decision of the Parliament of the Federal Republic of Germany via the Federal Office for Agriculture and Food (BLE). The funders had no role in the study design, data collection, analysis, or the decision to publish. Great thanks to district officials and village leaders in both Ruangwa and Nachingwea districts for their support and willingness to participate in the study and for providing the requested information.

**Conflicts of Interest:** The authors of this article declare no conflicts of interest. The funders had no role in the design, execution, interpretation, or writing of the study.

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
