# Peer review of "Preferences and Consumption of Pigeon Peas among Rural Households as Determinants for Developing Diversified Products for Sustainable Health"

_sustainability, doi:10.3390/su12156130_

Round 1
Reviewer 1 Report
The manuscript aimed to analyse existing recipes and consumption patterns related to consumer preferences and to determine the nutritional knowledge and actual perception of pigeon peas consumption, is original, very interesting and well described.
Author Response
Spell has been checked

Reviewer 2 Report
Dear Authors,
In the following I have some comments to improve the quality of the manuscript:
- There is not any mentioning of scientific name of “pigeon pea” (Cajanus cajan L.), I would suggest adding it with the family name
- L192: please define “PP” for the first use, check the whole text in case of the used abbreviations
- In Table 6, please check and correct the digits, e.g. “.638” to “0.638”, etc.
- The “Conclusion” section is too long, please make it brief
- Please check the references again which must be according to the “Author’s guidelines”
Author Response
The comments have been addressed.
